# SEED: Self-supervised Distillation For Visual Representation

**Zhiyuan Fang**[†] , **Jianfeng Wang**[‡]**, Lijuan Wang**[‡]**, Lei Zhang**[‡]**, Yezhou Yang**[†]**, Zicheng Liu**[‡]

[†]Arizona State University,          [‡]Microsoft Corporation

{zy.fang, yz.yang}@asu.edu
{jianfw, lijuanw, leizhang, zliu}@microsoft.com

## ABSTRACT

This paper is concerned with self-supervised learning for small models. The problem is motivated by our empirical studies that while the widely used contrastive self-supervised learning method has shown great progress on large model training, it does not work well for small models. To address this problem, we propose a new learning paradigm, named **SE**lf-Sup**E**rvised **D**istillation (SEED), where we leverage a larger network (as Teacher) to transfer its representational knowledge into a smaller architecture (as Student) in a self-supervised fashion. Instead of directly learning from unlabeled data, we train a student encoder to mimic the similarity score distribution inferred by a teacher over a set of instances. We show that SEED dramatically boosts the performance of small networks on downstream tasks. Compared with self-supervised baselines, SEED improves the top-1 accuracy from 42.2% to 67.6% on EfficientNet-B0 and from 36.3% to 68.2% on MobileNet-V3-Large on the ImageNet-1k dataset.

## 1 INTRODUCTION

The burgeoning studies and success on self-supervised learning (*SSL*) for visual representation are mainly marked by its extraordinary potency of learning from unlabeled data at scale. Accompanying with the *SSL* is its phenomenal benefit of obtaining task-agnostic representations while allowing the training to dispense with prohibitively expensive data labeling. Major ramifications of visual *SSL* include pretext tasks (Noroozi & Favaro, 2016; Zhang et al., 2016; Gidaris et al., 2018; Zhang et al., 2019; Feng et al., 2019), contrastive representation learning (Wu et al., 2018; He et al., 2020; Chen et al., 2020a), online/offline clustering (Yang et al., 2016; Caron et al., 2018; Li et al., 2020; Caron et al., 2020; Grill et al., 2020), etc. Among them, several recent works (He et al., 2020; Chen et al., 2020a; Caron et al., 2020) have achieved comparable or even better accuracy than the supervised pre-training when transferring to downstream tasks, *e.g.* semi-supervised classification, object detection.

The aforementioned top-performing *SSL* algorithms all involve large networks (*e.g.*, ResNet-50 (He et al., 2016) or larger), with, however, little attention on small networks. Empirically, we find that existing

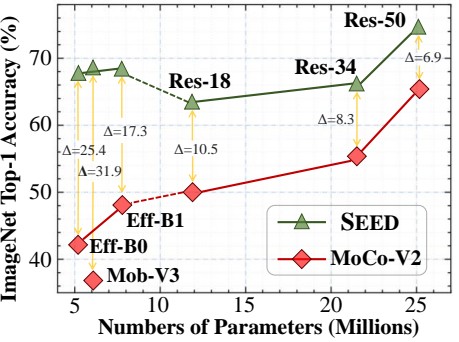

**Figure 1:** SEED vs. MoCo-V2 (Chen et al., 2020c)) on ImageNet-1K linear probe accuracy. The vertical axis is the top-1 accuracy and the horizontal axis is the number of learnable parameters for different network architectures. Directly applying self-supervised contrastive learning (MoCo-V2) does not work well for smaller architectures, while our method (SEED) leads to dramatic performance boost. Details of the setting can be found in Section 4.

techniques like contrastive learning do not work well on small networks. For instance, the linear probe top-1 accuracy on ImageNet using MoCo-V2 (Chen et al., 2020c) is only 36.3% with MobileNet-V3-Large (see Figure 1), which is much lower compared with its supervised training accuracy

75.2% (Howard et al., 2019). For EfficientNet-B0, the accuracy is 42.2% compared with its supervised training accuracy 77.1% (Tan & Le, 2019). We conjecture that this is because smaller models with fewer parameters cannot effectively learn instance level discriminative representation with large amount of data.

To address this challenge, we inject knowledge distillation (KD) (Buciluǎ et al., 2006; Hinton et al., 2015) into self-supervised learning and propose self-supervised distillation (dubbed as SEED) as a new learning paradigm. That is, train the larger, and distill to the smaller both in self-supervised manner. Instead of directly conducting self-supervised training on a smaller model, SEED first trains a large model (as the teacher) in a self-supervised way, and then distills the knowledge to the smaller model (as the student). Note that the conventional distillation is for supervised learning, while the distillation here is in the self-supervised setting without any labeled data. Supervised distillation can be formulated as training a student to mimic the probability mass function over classes predicted by a teacher model. In unsupervised knowledge distillation setting, however, the distribution over classes is not directly attainable. Therefore, we propose a simple yet effective self-supervised distillation method. Similar to (He et al., 2020; Wu et al., 2018), we maintain a queue of data samples. Given an instance, we first use the teacher network to obtain its similarity scores with all the data samples in the queue as well as the instance itself. Then the student encoder is trained to mimic the similarity score distribution inferred by the teacher over these data samples.

The simplicity and flexibility that SEED brings are self-evident. 1) It does not require any clustering/prototypical computing procedure to retrieve the pseudo-labels or latent classes. 2) The teacher model can be pre-trained with any advanced *SSL* approach, *e.g.*, MoCo-V2 (Chen et al., 2020c), SimCLR (Chen et al., 2020a), SWAV (Caron et al., 2020). 3) The knowledge can be distilled to any target small networks (either shallower, thinner, or totally different architectures).

To demonstrate the effectiveness, we comprehensively evaluate the learned representations on series of downstream tasks, *e.g.*, fully/semi-supervised classification, object detection, and also assess the transferability to other domains. For example, on ImageNet-1k dataset, SEED improves the linear probe accuracy of EfficientNet-B0 from 42.2% to 67.6% (a gain over 25%), and MobileNet-V3 from 36.3% to 68.2% (a gain over 31%) compared to MoCo-V2 baselines, as shown in Figure 1 and Section 4.

Our contributions can be summarized as follows:

- We are the first to address the problem of self-supervised visual representation learning for small models.

- We propose a self-supervised distillation (SEED) technique to transfer knowledge from a large model to a small model without any labeled data.

- With the proposed distillation technique (SEED), we significantly improve the state-of-the-art *SSL* performance on small models.

- We exhaustively compare a variety of distillation strategies to show the validity of SEED under multiple settings.

## 2 RELATED WORK

Among the recent literature in self-supervised learning, contrastive based approaches show prominent results on downstream tasks. Majority of the techniques along this direction are stemming from noise-contrastive estimation (Gutmann & Hyvärinen, 2010) where the latent distribution is estimated by contrasting with randomly or artificially generated noises. Oord et al. (2018) first proposed Info-NCE to learn image representations by predicting the future using an auto-regressive model for unsupervised learning. Follow-up works include improving the efficiency (Hénaff et al., 2019), and using multi-view as positive samples (Tian et al., 2019b). As these approaches can only have the access to limited negative instances, Wu et al. (2018) designed a memory-bank to store the previously seen random representations as negative samples, and treat each of them as independent categories (instance discrimination). However, this approach also comes with a deficiency that the previously stored vectors are inconsistent with the recently computed representations during the earlier stage of pre-training. Chen et al. (2020a) mitigate this issue by sampling negative samples from a large batch. Concurrently, He et al. (2020) improve the memory-bank based method and propose to use

the momentum updated encoder for the remission of representation inconsistency. Other techniques include Misra & Maaten (2020) that combines the pretext-invariant objective loss with contrastive learning, and Wang & Isola (2020) that decomposes contrastive loss into alignment and uniformity objectiveness.

Knowledge distillation (Hinton et al., 2015) aims to transfer knowledge from a cumbersome model to a smaller one without losing too much generalization power, which is also well investigated in model compression (Buciluǎ et al., 2006). Instead of mimicking the teacher's output logit, attention transfer (Zagoruyko & Komodakis, 2016) formulates knowledge distillation on attention maps. Similarly, works in (Ahn et al., 2019; Yim et al., 2017; Koratana et al., 2019; Huang & Wang, 2017) have utilized different learning objectives including consistency on feature maps, consistency on probability mass function, and maximizing the mutual information. CRD (Tian et al., 2019a), which is derived from CMC (Tian et al., 2019b), optimizes the student network by a similar objective to Oord et al. (2018) using a derived lower bound on mutual information. However, the aforementioned efforts all focus on task-specific distillation (*e.g.*, image classification) during the fine-tuning phase rather than a task-agnostic distillation in the pre-training phase for the representation learning. Several works on natural language pre-training proposed to leverage knowledge distillation for a smaller yet stronger small models. For instances, DistillBert (Sanh et al., 2019), TinyBert (Jiao et al., 2019), and MobileBert (Sun et al., 2020), have used knowledge distillation for model compression and shown their validity on multiple downstream tasks. Similar works also emphasize the value of smaller and faster models for language representation learning by leveraging knowledge distillation (Turc et al., 2019; Sun et al., 2019). These works all demonstrate the effectiveness of knowledge distillation for language representation learning in small models, while are not extended to the pre-training for visual representations. Notably, a recent concurrent work CompRess (Abbasi Koohpayegani et al., 2020) also point out the importance to develop better *SSL* method for smaller models. SEED closely relates to the above techniques but aims to facilitate **visual representation learning during pre-training phase using distillation technique for small models**, which as far as we know has not yet been investigated.

## 3 METHOD

### 3.1 PRELIMINARY ON KNOWLEDGE DISTILLATION

Knowledge distillation (Hinton et al., 2015; Buciluǎ et al., 2006) is an effective technique to transfer knowledge from a strong teacher network to a target student network. The training task can be generalized as the following formulation:

$$\hat{\theta}_S = \arg\min_{\theta_S} \sum_i^N \mathcal{L}_{\text{sup}}(\mathbf{x}_i, \theta_S, y_i) + \mathcal{L}_{\text{distill}}(\mathbf{x}_i, \theta_S, \theta_T), \qquad (1)$$

where $\mathbf{x}_i$ is an image, $y_i$ is the corresponding annotation, $\theta_S$ is the parameter set for the student network, and $\theta_T$ is the set for the teacher network. The loss $\mathcal{L}_{\text{sup}}$ is the alignment error between the network prediction and the annotation. For example in image classification task (Mishra & Marr, 2017; Shen & Savvides, 2020; Polino et al., 2018; Cho & Hariharan, 2019), it is normally a cross entropy loss. For object detection (Liu et al., 2019; Chen et al., 2017), it includes bounding box regression as well. The loss of $\mathcal{L}_{\text{distill}}$ is the mimic error of the student network towards a pre-trained teacher network. For example in (Hinton et al., 2015), the teacher signal comes from the softmax prediction of multiple large-scale networks and the loss is measured by the *Kullback–Leibler* divergence. In Romero et al. (2014), the task is to align the intermediate feature map values and to minimize the squared $l2$ distance. The effectiveness has been well demonstrated in the supervised setting with labeled data, but remains unknown for the unsupervised setting, which is our focus.

### 3.2 SELF-SUPERVISED DISTILLATION FOR VISUAL REPRESENTATION

Different from supervised distillation, SEED aims to transfer knowledge from a large model to a small model without requiring labeled data, so that the learned representations in small model can be used for downstream tasks. Inspired by contrastive *SSL*, we formulate a simple approach for the distillation on the basis of instance similarity distribution over a contrastive instance queue. Similar to He et al. (2020), we maintain an instance queue for storing data samples' encoding output from the

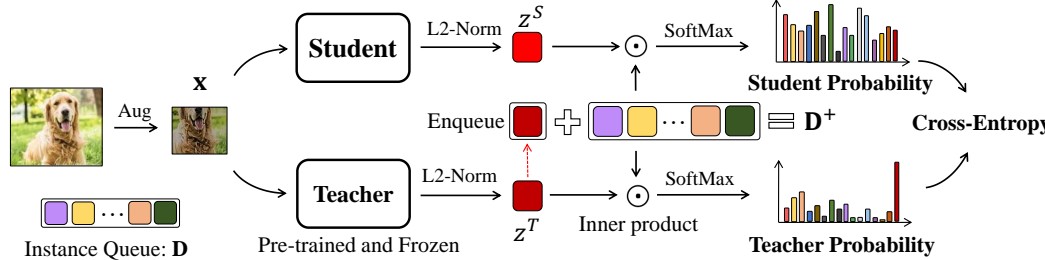

**Figure 2:** Illustration of our self-supervised distillation pipeline. The teacher encoder is pre-trained by *SSL* and kept frozen during the distillation. The student encoder is trained by minimizing the cross entropy of probabilities from teacher & student for an augmented view of an image, computed over a dynamically maintained queue.

teacher. Given a new sample, we compute its similarity scores with all the samples in the queue using both the teacher and the student models. We require that the similarity score distribution computed by the student matches with that computed by the teacher, which is formulated as minimizing the cross entropy between the student and the teacher's similarity score distributions (as illustrated in Figure 2).

Specifically, for a randomly augmented view $\mathbf{x}_i$ of an image, it is first mapped and normalized into feature vector representations $\mathbf{z}_i^T = f_\theta^T(\mathbf{x}_i)/\|f_\theta^T(\mathbf{x}_i)\|_2$, and $\mathbf{z}_i^S = f_\theta^S(\mathbf{x}_i)/\|f_\theta^S(\mathbf{x}_i)\|_2$, where $\mathbf{z}_i^T, \mathbf{z}_i^S \in \mathbb{R}^D$, and $f_\theta^T$ and $f_\theta^S$ denote the teacher and student encoders, respectively. Let $\mathbf{D} = [\mathbf{d}_1...\mathbf{d}_K]$ denote the instance queue where $K$ is the queue length and $\mathbf{d}_j$ is the feature vector obtained from the teacher encoder. Similar to the contrastive learning framework, $\mathbf{D}$ is progressively updated under the "*first-in first-out*" strategy as distillation proceeds. That is, we en-queue the visual features of the current batch inferred by the teacher and de-queue the earliest seen samples at the end of iteration. Note that the maintained samples in queue $\mathbf{D}$ are mostly random and irrelevant to the target instance $\mathbf{x}_i$. Minimizing the cross entropy between the similarity score distribution computed by the student and teacher based on $\mathbf{D}$ softly contrasts $\mathbf{x}_i$ with randomly selected samples, without directly aligning with the teacher encoder. To address this problem, we add the teacher's embedding $(\mathbf{z}_i^T)$ into the queue and form $\mathbf{D}^+ = [\mathbf{d}_1...\mathbf{d}_K, \mathbf{d}_{K+1}]$ with $\mathbf{d}_{K+1} = \mathbf{z}_i^T$.

Let $\mathbf{p}^T(\mathbf{x}_i; \theta_T; \mathbf{D}^+)$ denote the similarity score between the extracted teacher feature $\mathbf{z}_i^T$ and $\mathbf{d}_j$'s $(j = 1, ..., K + 1)$ computed by the teacher model. $\mathbf{p}^T(\mathbf{x}_i; \theta_T; \mathbf{D}^+)$ is defined as

$$\mathbf{p}^T(\mathbf{x}_i; \theta_T, \mathbf{D}^+) = \begin{bmatrix} p_1^T ... p_{K+1}^T \end{bmatrix}, \qquad p_j^T = \frac{\exp(\mathbf{z}_i^T \cdot \mathbf{d}_j/\tau^T)}{\sum_{\mathbf{d} \sim \mathbf{D}^+} \exp(\mathbf{z}_i^T \cdot \mathbf{d}/\tau^T)}, \qquad (2)$$

and $\tau^T$ is a temperature parameter for the teacher. Note, we use $()^T$ to represent the feature from the teacher network and use $(\cdot)$ to represent the inner product between two features.

Similarly let $\mathbf{p}^S(x_i; \theta_S, \mathbf{D}^+)$ denote the similarity score computed by the student model, which is defined as

$$\mathbf{p}^S(\mathbf{x}_i; \theta_S, \mathbf{D}^+) = \begin{bmatrix} p_1^S ... p_{K+1}^S \end{bmatrix}, \qquad \text{where } p_j^S = \frac{\exp(\mathbf{z}_i^S \cdot \mathbf{d}_j/\tau^S)}{\sum_{\mathbf{d} \sim \mathbf{D}^+} \exp(\mathbf{z}_i^S \cdot \mathbf{d}/\tau^S)}, \qquad (3)$$

and $\tau^S$ is a temperature parameter for the student.

Our self-supervised distillation can be formulated as minimizing the cross entropy between the similarity scores of the teacher, $\mathbf{p}^T(\mathbf{x}_i; \theta_T, \mathbf{D}^+)$, and the student, $\mathbf{p}^S(\mathbf{x}_i; \theta_S, \mathbf{D}^+)$, over all the instances $\mathbf{x}_i$, that is,

$$\hat{\theta}_S = \arg\min_{\theta_S} \sum_i^N -\mathbf{p}^T(\mathbf{x}_i; \theta_T, \mathbf{D}^+) \cdot \log \mathbf{p}^S(\mathbf{x}_i; \theta_S, \mathbf{D}^+)$$

$$= \arg\min_{\theta_S} \sum_i^N \sum_j^{K+1} -\frac{\exp(\mathbf{z}_i^T \cdot \mathbf{d}_j/\tau^T)}{\sum_{\mathbf{d} \sim \mathbf{D}^+} \exp(\mathbf{z}_i^T \cdot \mathbf{d}/\tau^T)} \cdot \log \frac{\exp(\mathbf{z}_i^S \cdot \mathbf{d}_j/\tau^S)}{\sum_{\mathbf{d} \sim \mathbf{D}^+} \exp(\mathbf{z}_i^S \cdot \mathbf{d}/\tau^S)}. \qquad (4)$$

Since the teacher network is pre-trained and frozen, the queued features are consistent during training *w.r.t.* the student network. The higher the value of $p_j^T$ is, the larger weight will be laid on $p_j^S$. Due

to the $l2$ normalization, similarity score between $\mathbf{z}_i^T$ and $\mathbf{d}_{K+1}$ remains constant $\mathbb{1}$ before softmax normalization, which is the largest among $p_j^T$. Thus, the weight for $p_{K+1}^T$ is the largest and can be adjusted solely by tuning the value of $\tau^T$. By minimizing the loss, the feature of $\mathbf{z}_i^S$ can be aligned with $\mathbf{z}_i^T$ and meanwhile contrasts with other unrelated image features in $\mathbf{D}$. We further discuss the relation of these two goals with our learning objective in Appendix A.5.

**Relations with *Info-NCE* loss.** When $\tau^T \to 0$, the softmax function for $\mathbf{p}^T$ smoothly approaches to a one-hot vector, where $p_{K+1}^T$ equals 1 and all others 0. In this extreme case, the loss becomes

$$\mathcal{L}_{NCE} = \sum_i^N -\log \frac{\exp(\mathbf{z}_i^T \cdot \mathbf{z}_i^S / \tau)}{\sum_{\mathbf{d} \sim \mathbf{D}^+} \exp(\mathbf{z}_i^S \cdot \mathbf{d} / \tau)}, \tag{5}$$

which is similar to the widely-used *Info-NCE* loss (Oord et al., 2018) in contrastive-based *SSL* (see discussion in Appendix A.6.

## 4 EXPERIMENT

### 4.1 PRE-TRAINING

**Self-Supervised Pre-training of Teacher Network.** By default, we use MoCo-V2 (Chen et al., 2020c) to pre-train the teacher network. Following (Chen et al., 2020a), we use ResNet as the network backbone with different depths/widths and append a multi-layer-perceptron (MLP) layer (two linear layers and one *ReLU* (Nair & Hinton, 2010) activation layer in between) at the end of the encoder after average pooling. The dimension of the last feature dimension is 128. All teacher networks are pre-trained for 200 epochs due to the computational limitation unless explicitly specified. As our distillation is independent with the teacher pre-training algorithm, we also show results with other self-supervised pre-trained models for teacher network, *e.g.*, SWAV (Caron et al., 2020), SimCLR (Chen et al., 2020a).

**Self-Supervised Distillation on Student Network.** We choose multiple smaller networks with fewer learnable parameters as the student network: MobileNet-v3-Large (Howard et al., 2017), EfficientNet-B0 (Tan & Le, 2019), and smaller ResNet with fewer layers (ResNet-18, 34). Similar to the pre-training for teacher network, we add one additional MLP layer on the basis of the student network. Our distillation is trained with a standard SGD optimizer with momentum 0.9 and a weight decay parameter of 1e-4 for 200 epochs. The initial learning rate is set as 0.03 and updated by a cosine decay scheduler (Nair & Hinton, 2010) with 5 warm-up epochs and batch size 256. In *Eq.* 4, the teacher temperature is set as $\tau^T = 0.01$ and the student temperature is $\tau^S = 0.2$. The queue size of $K$ is 65,536. In the following subsections and appendix, we also show results with different hyper-parameter values, *e.g.*, for $\tau^T$ and $K$.

### 4.2 FINE-TUNING AND EVALUATION

In order to validate the effectiveness of self-supervised distillation, we choose to assess the performance of representations of the student encoder on several downstream tasks. We first report its performances of linear evaluation and semi-supervised linear evaluation on the ImageNet ILSVRC-2012 (Deng et al., 2009) dataset. To measure the feature transferability brought by distillation, we also conduct evaluations on other tasks, which include object detection and segmentation on the VOC07 (Everingham et al.) and MS-COCO (Lin et al., 2014) datasets. At the end, we compare the transferability of the features learned by distillation with ordinary self-supervised contrastive learning on the tasks of linear classification on datasets from different domains.

**Linear and *K*NN Evaluation on ImageNet.** We conduct the supervised linear classification on ImageNet-1K, which contains $\sim$1.3M images for training, and 50,000 images for validation, spanning 1,000 categories. Following previous works in (He et al., 2020; Chen et al., 2020a), we train a single linear layer classifier on top of the frozen network encoder after self-supervised pre-training/distillation. SGD optimizer is used to train the linear classifier for 100 epochs with weight decay to be 0. The initial learning rate is set as 30 and is then reduced by a factor of 10 at 60 and 80 epochs (similar as in Tian et al. (2019a)). Notably, when training the linear classifier for MobileNet and EfficientNet, we reduce the initial learning rate to 3. The results are reported in terms of Top-1

**Table 1:** ImageNet-1k test *accuracy* (%) using $K$NN and linear classification for multiple students and MoCo-v2 pre-trained *deeper* teacher architectures. ✗ denotes MoCo-V2 self-supervised learning baselines before distillation. * indicates using a deeper teacher encoder pre-trained by SWAV, where additional small-patches are also utilized during distillation and trained for 800 epochs. $K$ denotes Top-1 accuracy using $K$NN. T-1 and T-5 denote Top-1 and Top-5 accuracy using linear evaluation. First column shows Top-1 Acc. of Teacher network. First row shows the supervised performances of student networks.

| S / T | T-1 | Eff-b0 $K$ | T-1 | T-5 | Eff-b1 $K$ | T-1 | T-5 | Mob-v3 $K$ | T-1 | T-5 | R-18 $K$ | T-1 | T-5 | R-34 $K$ | T-1 | T-5 |
|---|---|---|---|---|---|---|---|---|---|---|---|---|---|---|---|---|
| **Supervised Acc.** | | | 77.3 | | | 79.2 | | | 75.2 | | | 72.1 | | | 75.0 | |
| ✗ | - | 30.0 | 42.2 | 68.5 | 34.4 | 50.7 | 74.6 | 27.5 | 36.3 | 62.2 | 36.7 | 52.5 | 77.0 | 41.5 | 57.4 | 81.6 |
| **R-50** $\triangle$ | 67.4 | 46.0 +16.0 | 61.3 +19.1 | 82.7 +14.2 | 46.1 +16.1 | 61.4 +10.7 | 83.1 +8.8 | 44.8 +17.3 | 55.2 +18.9 | 80.3 +18.1 | 43.4 +6.7 | 57.9 +5.1 | 82.0 +4.8 | 45.2 +3.7 | 58.5 +1.1 | 82.6 +1.0 |
| **R-101** $\triangle$ | 70.3 | 50.1 +20.1 | 63.0 +20.8 | 83.8 +15.3 | 50.3 +15.9 | 63.4 +12.7 | 84.6 +10.0 | 48.8 +21.3 | 59.9 +23.6 | 83.5 +21.3 | 48.6 +11.9 | 58.9 +6.4 | 82.5 +5.5 | 50.5 +9.0 | 61.6 +4.2 | 84.9 +3.3 |
| **R-152** $\triangle$ | 74.2 | 50.7 +20.7 | 65.3 +23.1 | 86.0 +17.5 | 52.4 +18.0 | 67.3 +16.6 | 86.9 +12.3 | 49.5 +22.0 | 61.4 +25.1 | 84.6 +22.4 | 49.1 +12.4 | 59.5 +7.0 | 83.3 +6.3 | 51.4 +9.9 | 62.7 +5.3 | 85.8 +4.2 |
| **R50×2*** $\triangle$ | 77.3 | 57.4 +27.4 | 67.6 +25.4 | 87.4 +18.9 | 60.3 +25.9 | 68.0 +17.3 | 87.6 +13.0 | 55.9 +18.9 | 68.2 +31.9 | 88.2 +26.0 | 55.3 +18.6 | 63.0 +10.5 | 84.9 +7.9 | 58.2 +16.7 | 65.7 +8.3 | 86.8 +5.2 |

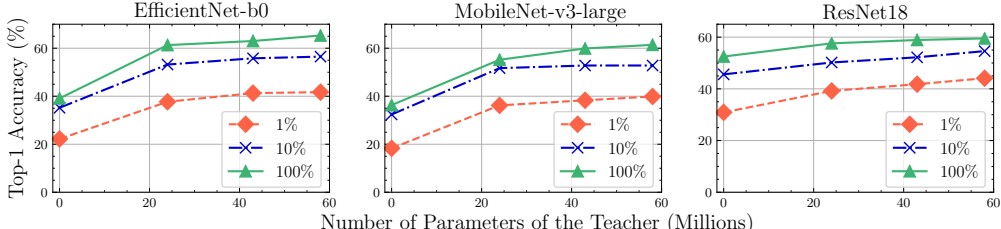

**Figure 3:** ImageNet-1k Top-1 *accuracy* for semi-supervised evaluations using 1% (red line), 10% (blue line) of the annotations for linear fine-tuning, in comparison with the fully supervised (green line) linear evaluation baseline for SEED. For the points whose Teacher's number of parameters is at 0, we show the semi-supervised linear evaluation results of MoCo-V2 without any distillation. The Student models tend to perform better on the semi-supervised tasks after distillation from larger Teachers.

and Top-5 accuracy. We also perform classification using $K$-Nearest Neighbors ($K$NN) based on the learned 128d vector from the last MLP layer. The sample is classified by taking the most frequent label of its $K$ ($K = 10$) nearest neighbors.

Table 1 shows the results with various teacher networks and student networks. We list the baseline of contrastive self-supervised pre-training using MoCo-V2 (Chen et al., 2020c) in the first row for each student architecture. We can see clearly that smaller networks perform rather worse. For example, MobileNet-V3 can only reach 36.3%. This aligns well with previous conclusions from (Chen et al., 2020a;b) that bigger models are desired to perform better in contrastive-based self-supervised pre-training. We conjecture that this is mainly caused by the inability of smaller network to discriminate instances in a large-scale dataset. The results also clearly demonstrate that the distillation from a larger network helps boosting the performances of small networks, and show obvious improvement. For instance, with MoCo-V2 pre-trained ResNet-152 (for 400 epochs) as the teacher network, the Top-1 accuracy of MobileNet-V3-Large can be significantly improved from 36.3% to 61.4%. Furthermore, we use ResNet-50×2 (provided in Caron et al. (2020)) as the teacher network and adopt the multi-crop trick (see A.2 for details). The accuracy can be further improved to 68.2% (last row of Table 1) for MobileNet-V3-Large with 800 epochs of distillation. We note that the gain benefited from distillation becomes more distinct on smaller architectures and we further study the effect of various teacher models in ablations.

**Semi-Supervised Evaluation on ImageNet.** Following (Oord et al., 2018; Kornblith et al., 2019; Kolesnikov et al., 2019), we evaluate the representation on the semi-supervised task, where a fixed 1% or 10% subsets of ImageNet training data (Chen et al., 2020a) are provided with the annotations. After the self-supervised learning with and without distillation, we also train a classifier on top of the representation. The results are shown in Figure 3, where the baseline without distillation is depicted

**Table 2:** Object detection and instance segmentation results using contrastive self-supervised learning and SEED distillation using ResNet-18 as backbone: bounding-box AP ($AP^{bb}$) and mask AP ($AP^{mk}$) evaluated on VOC07-val and COCO testing split. More results on different backbones can be found in the Appendix. Subscript in green represents improvement is larger than 0.3.

| S | T | VOC Obj. Det. | | | COCO Obj. Det. | | | COCO Inst. Segm. | | |
|---|---|---|---|---|---|---|---|---|---|---|
| | | $AP^{bb}$ | $AP^{bb}_{50}$ | $AP^{bb}_{75}$ | $AP^{bb}$ | $AP^{bb}_{50}$ | $AP^{bb}_{75}$ | $AP^{mk}$ | $AP^{mk}_{50}$ | $AP^{mk}_{75}$ |
| | ✗ | 46.1 | 74.5 | 48.6 | 35.0 | 53.9 | 37.7 | 31.0 | 51.1 | 33.1 |
| | R-50 | 46.1(0.0) | 74.8(+0.3) | 49.1(+0.5) | 35.3(+0.3) | 54.2(+0.3) | 37.8(+0.1) | 31.1(+0.1) | 51.1(0.0) | 33.2(+0.1) |
| R-18 | R-101 | 46.8(+0.7) | 75.8(+1.3) | 49.3(+0.7) | 35.3(+0.3) | 54.3(+0.4) | 37.9(+0.2) | 31.3(+0.3) | 51.3(+0.2) | 33.4(+0.3) |
| | R-152 | 46.8(+0.7) | 75.9(+1.4) | 50.2(+1.6) | 35.4(+0.4) | 54.4(+0.5) | 38.0(+0.3) | 31.3(+0.3) | 51.4(+0.3) | 33.4(+0.3) |

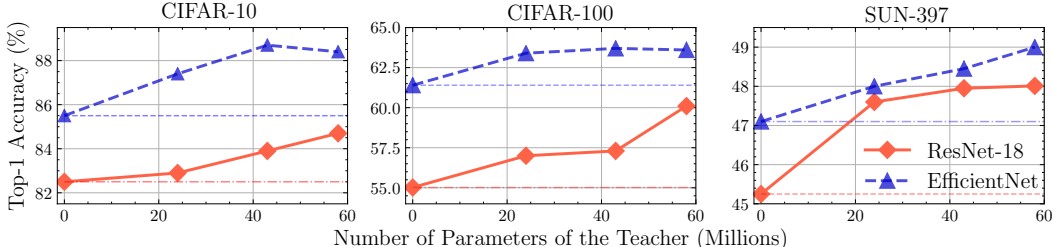

**Figure 4:** ImageNet-1k *Accuracy* (%) of student network (EfficientNet-B0 and ResNet-18) transferred to other domains (CIFAR-10, CIFAR-100, SUN-397 datasets) with and without distillation from lager architectures (ResNet-50/101/152).

when teacher parameters are 0. As we can see, the accuracy is also improved remarkably with SEED distillation, and a stronger teacher network with more parameters leads to a better performed student network.

**Transferring to Classification.** To further study whether the improvement of the learned representations by distillation is confined to ImageNet, we evaluate on additional classification datasets to study the generalization and transferability of the feature representation. We strictly follow the linear evaluation and fine-tuning settings from (Kornblith et al., 2019; Chen et al., 2020a; Grill et al., 2020), that a linear layer is trained on the basis of frozen features. We report Top-1 Accuracy of models before and after distillation from various architectures on CIFAR-10, CIFAR-100 (Krizhevsky et al., 2009), SUN-397 (Xiao et al., 2010) datasets (see Figure 4). More details regarding pre-processing and training can be found in A.1.2. Notably, we observe that our distillation surpasses contrastive self-supervised pre-training consistently on all benchmarks, verifying the effectiveness of SEED. This also proves the generalization ability of the learned representations from distillation to a wide range of data domain and different classes.

**Transferring to Detection and Segmentation.** We conduct two downstream tasks here. The first is Faster R-CNN (Ren et al., 2015) model for object detection trained on VOC-07+12 train+val set and evaluated on VOC-07 test split. The second is Mask R-CNN (He et al., 2017) model for the object detection and instance segmentation on COCO 2017 dataset (Lin et al., 2014). The pre-trained model serves as the initial weight and following He et al. (2020), we fine-tune all the layers of the model. More experiment settings can be found in A.2. The results are illustrated in Table 2. As we can see, on VOC, the distilled pre-trained model achieves a large improvement. With ResNet-152 as the teacher network, the Resnet18-based Faster R-CNN model shows +0.7 point improvement on AP, +1.4 improvement on $AP_{50}$ and +1.6 on $AP_{75}$. On COCO, the improvement is relatively minor and the reason could be that COCO training set has ∼118k training images while VOC has only ∼16.5k training images. A larger training set with more fine-tuning iterations reduces the importance of the initial weights.

## 4.3 ABLATION STUDY

We now explore the effects of distillation using different Teacher architectures, Teacher Pre-training algorithms, various distillation strategies and hyper-parameters.

**Table 3:** ImageNet-1k *Accuracy* (%) of student network (ResNet-18) distilled from variants of self-supervised ResNet-50. P-E/D-E represent the pretraining and distillation epochs. T./S.-Top represent testing accuracy of Teacher and Student. * represents distillation using additional small patches. First row is the ResNet-18 *SSL* baseline using MoCo-v2 trained for 200 epochs.

| Teacher | P-E | D-E | T. Top-1 | S. Top-1 | S. Top-5 |
|---------|-----|-----|----------|----------|----------|
| ✗ | ✗ | ✗ | ✗ | 52.5 | 77.0 |
| MoCo | 200 | 200 | 60.6 | 52.1 | 77.0 |
| SimCLR | 200 | 200 | 65.6 | 57.5 | 81.7 |
| MoCo-v2 | 200 | 200 | 67.4 | 57.9 | 82.0 |
| | 800 | 200 | 71.1 | 60.5 | 83.5 |
| SWAV | 800 | 100 | 75.3 | 61.1 | 83.8 |
| | 800 | 200 | 75.3 | 61.7 | 84.2 |
| | 800 | 400 | 75.3 | 62.0 | 84.4 |
| SWAV* | 800 | 200 | 75.3 | **62.6** | **84.8** |

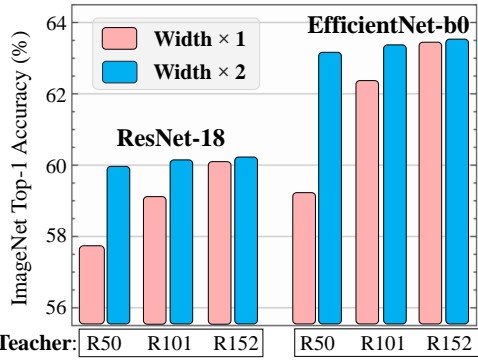

**Figure 5:** *Accuracy* (%) of student networks (EfficientNet-b0 and ResNet-18) on ImageNet distilled from *wider* MoCo-v2 pre-trained ResNet (ResNet-50/101/152×2).

**Different Teacher Networks.** Figure 5 summarizes the accuracy of ResNet-18 and EfficientNet-B0 distilled from wider and deeper ResNet architectures. We see clear performance improvement as depth and width of teacher network increase: compared to ResNet-50, deeper (ResNet-101) and wider (ResNet-50×2) substantially improve the accuracy. However, further architectural enlargement has relatively limited effects, and we suspect the accuracy might be limited by the student network capacity in this case.

**Different Teacher Pre-training Algorithms.** In Table 3, we show the Top-1 accuracy of ResNet-18 distilled from ResNet-50 with different pre-training algorithms, *i.e.*, MoCo-V1 (He et al., 2020), MoCo-V2 (Chen et al., 2020c), SimCLR (Chen et al., 2020a), and SWAV (Caron et al., 2020)). Notably, the aforementioned methods all unanimously adopt contrastive-based pre-training except SWAV, which is based upon online clustering. We find that our SEED is agnostic to pre-training approaches, making it easy to use any self-supervised models (including clustering-based approach like SWAV) in self-supervised distillation. In addition, we observe that more training epochs for both teacher *SSL* and distillation epochs can bring beneficial gain.

**Other Distillation Strategies.** We explore several alternative distillation strategies. *l2-Distance*: where the *l2*-distance of teacher & student's embeddings are minimized, motivated by Romero et al. (2014). *K-Means*: we exploit *K*-Means clustering to assign a pseudo-label based on the teacher network's representation. *Online Clustering*: we continuously update the clustering centers during distillation for pseudo-label generation. *Binary Contrastive Loss*: we adopt an Info-NCE alike loss for contrastive distillation (Tian et al., 2019a). We provide details for other strategies in A.4. Table 4 shows the results for each method on ResNet-18 (student) distilled from ResNet-50. From the results, the simple *l2*-distance minimizing approach can achieve a decent accuracy, which demonstrates the effectiveness of applying the distillation idea to the self-supervised learning. Beyond that, we study the effect of the original *SSL* (MoCo-V2) supervision as supplementary loss to SEED and find it does not bring additional benefits to distillation. We find close results from these two strategies (Top-1 linear Acc.), SEED achieves 57.9%, while SEED + MoCo-V2 achieves 57.6%. This implies that the loss of SEED can to a large extent cover the original *SSL* loss, and it is not necessary to conduct *SSL* any further during distillation. Meanwhile, our proposed SEED outperforms these alternatives with highest accuracy, which shows the superiority of aligning the student towards the teacher and contrasting with the irrelevant samples.

**Other Hyper-Parameters.** Table 5 summarizes the distillation performances on multiple datasets using different temperature $\tau^T$. We observe a better performance when decreasing $\tau^T$ to 0.01 for ImageNet-1k and CIFAR-10 dataset, and to 1e-3 for CIFAR-100 datasets. When $\tau$ is large, the softmax-normalized similarity score of $p_j^T$ between $\mathbf{z}_i^T$ and instance $\mathbf{d}_j$ in the queue $\mathbf{D}^+$ also becomes large, which means the student's feature should be less discriminative with the features of other images to some extent. When $\tau^T$ is 0, the teacher model will generate a one-hot vector, which only treats $\mathbf{z}_i^T$ as a positive instance and all others in the queue as negative. Thus, the best $\tau$ is a trade-off depending on the data distribution. We further compare effect of different hyper-parameters in A.8.

**Table 4:** Top-1/5 accuracy of linear classification results on ImageNet using different distillation strategies on ResNet-18 (student) and ResNet-50 (teacher) architectures.

| Method | Top-1 Acc. | Top-5 Acc. |
|---|---|---|
| *l2-Distance* | 55.3 | 80.3 |
| *K*-Means | 51.0 | 75.8 |
| Online Clustering | 56.4 | 81.2 |
| Binary Contr. Loss | 57.4 | 81.5 |
| SEED + MoCo-V2 | 57.6 | 81.8 |
| SEED | **57.9** | **82.0** |

**Table 5:** Effect of $\tau^T$ for the distillation of ResNet-18 (student), ResNet-50 (teacher) on multiple datasets.

| $\tau^T$ | ImageNet | | CIFAR-10 | CIFAR-100 |
|---|---|---|---|---|
| | Top-1 | Top-5 | Top-1 | Top-1 |
| 0.3 | 54.8 | 80.0 | 78.7 | 46.6 |
| 0.1 | 54.9 | 80.1 | 83.0 | 50.1 |
| 0.05 | 56.5 | 81.3 | 84.4 | 56.2 |
| 0.01 | **57.9** | **82.0** | **87.5** | 60.6 |
| 1e-3 | 57.6 | 81.8 | 86.9 | **60.8** |

## 5 CONCLUSIONS

Self-Supervised Learning is acknowledged for its remarkable ability in learning from unlabeled, and large scale data. However, a critical impedance for the *SSL* pre-training on smaller architecture comes from its low capacity of discriminating enormous number of instances. Instead of directly learning from unlabeled data, we proposed SEED as a novel self-supervised learning paradigm, which learns representation by self-supervised distillation from a bigger *SSL* pre-trained model. We show in extensive experiments that SEED effectively addresses the weakness of self-supervised learning for small models and achieves state-of-the-art results on various benchmarks of small architectures.

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

## A    APPENDIX

We discuss more details and different hyperparameters for SEED during distillation.

### A.1    PSEUDO-IMPLEMENTATIONS

We provide pseudo-code of the SEED distillation in *Py*Torch Paszke et al. (2019) style:

```
1  '''Q: maintaining queue of previous representations: (N X D)
2  T: Cumbersome encoder as Teacher.
3  S: Target encoder as Student.
4  temp_T, temp_S: temperatures of the Teacher & Student.
5  '''
6
7  # activate evaluation mode for Teacher to freeze BN and updation.
8  T.eval()
9
10 for images in enumerate(loader): # Enumerate single crop-view
11
12     # augment image to get one identical view
13     images = aug(images)
14
15     # Batch-size
16     B = images.shape[0]
17
18     # extract embedding from S: 1 X D
19     X_S = S(images)
20     X_S = torch.norm(X_S, p=2, dim=1)
21
22     # use the gradient-free mode
23     with torch.no_grad():
24         X_T = T(image) # embedding from T: 1 X D
25         X_T = torch.norm(X_T, p=2, dim=1)
26
27     # insert the current batch embedding from T
28     enqueue(Q, X_T)
29
30     # probability scores distribution for T, S: B X (N + 1)
31     S_Dist = torch.einsum('bd, dn -> bn', [X_S], Q.t().clone().detach())
32     T_Dist = torch.einsum('bd, dn -> bn', [X_T], Q.t().clone().detach())
33
34     # Apply temperatures for soft-labels
35     S_Dist /= temp_S
36     T_Dist = SoftMax(T_Dist/temp_T, dim=1)
37
38     # loss computation, use log_softmax for stable computation
39     loss = -torch.mul(T_Dist, Log_SoftMax(S_Dist, dim=1)).sum()/B
40
41     # update the random sample queue
42     dequeue(Q, B)    # pop-out earliest B instances
43
44     # SGD updation
45     loss.backward()
46     update(S.params)
```

### A.1.1 Data Augmentations

Both our teacher pre-training and distillation adopt the data augmentations as follows:

> *Random Resized Crop*: The image is randomly resized with a scale of {0.2, 1.0}, then cropped to the size of 224×224.
>
> *Random Color Jittering*: with brightness to be {0.4, 0.4, 0.4, 0.1} with probability at 0.8.
>
> *Random Gray Scale* transformation: with probability at 0.2.
>
> *Random Gaussian Blur* transformation: with $\sigma$ = {0.1, 0.2} and probability at 0.5.
>
> *Horizontal Flip*: Horizontal flip is applied with probability at 0.5.

### A.1.2 Pre-training and Distillation on MobileNet and EfficientNet

MobileNet (Howard et al., 2017) and EfficientNet (Tan & Le, 2019) have been considered as the smaller counterparts with larger models, *i.e.*, ResNet-50 (with supervised training, EfficientNet-B0 hits 77.2% Top-1 Acc., and MobileNet-V3-large reaches 72.2% on ImageNet testing split). Nevertheless, un-matched performances are observed in the task of self-supervised contrastive pre-training: *i.e.*, Self-Supervised Learning (MoCo-V2) on MobileNet-V3 only yields 36.3% Top-1 Acc. on ImageNet. We conjecture that several reasons might lead to this dilemma:

1. The inability of models with less parameters for handling large volume of categories and data, which exists also in other domains, *i.e.*, face recognition (Guo et al., 2016; Zhang et al., 2017).

2. Less possibility for optimum parameters to be chosen when transferring to downstream tasks: models with more parameters after pre-training might produce a plenty cornucopia of optimum parameters for fine-tuning.

To narrow the dramatic performance gap between smaller architectures using contrastive *SSL* with the larger, we explore with architectural manipulations and training hyper-parameters. In specific, we find that by adding a deeper projection head largely improves the representation quality, *a.k.a.*, better performances on linear evaluation. We experiment with adding one additional linear projection head on the top of convolutional backbones.

Similarly, we also expand the MLP projection head on EfficientNet-b0. Though recent work shows that fine-tuning from a middle layer of the projection head can produce a largely different result (Chen et al., 2020b), we consistently just use the representations from convolutional trunk **without adding extra layers** during the phase of linear evaluation. As shown in Table 6, pre-training with a deeper projection head dramatically helps the improvement on linear evaluations, adding 17% Top-1. Acc. for Mobile-v3-large, and we report the improved baselines in the main paper (see the first row in Table 1 of the main paper). We keep most of the hyper-parameters as the distillation on ResNet except reducing the weight-decay of them to 1e-5, following (Tan & Le, 2019; Sandler et al., 2018).

**Table 6:** Linear evaluations on ImageNet of EfficientNet and MobileNet pre-trained using MoCo-v2. A deeper projection head largely boosts the linear evaluation performances on smaller architectures.

| Model | Deeper MLPs | Top-1 Acc. | Top-5 Acc. |
|---|---|---|---|
| EfficientNet-b0 | ✗ | 39.1 | 64.6 |
| EfficientNet-b0 | ✓ | 42.2 | 68.5 |
| Mobile-v3-large | ✗ | 19.0 | 41.3 |
| Mobile-v3-large | ✓ | 36.3 | 62.2 |

### A.2 Additional Details of Evaluations

We list additional details regarding our evaluation experiments in this section.

**ImageNet-1k Semi-Supervised Linear Evaluation.** Following Zhai et al. (2019); Chen et al. (2020a), we train the FC layers on the basis of our student encoder after distillation using a fraction

**Table 7:** Before and after distillation Top-1/5 test *accuracy* (%) on ImageNet of EfficientNet-b0 and MobileNet-large without deeper MLPs.

| Student | Teacher | Top-1 | Top-5 |
|---|---|---|---|
| EfficientNet-b0 | ✗ | 39.1 | 64.6 |
| | ResNet-50 | 59.2 | 81.2 |
| | ResNet-101 | 62.8 | 84.7 |
| | ResNet-152 | 63.3 | 85.6 |
| MobileNet-v3 | ✗ | 19.0 | 41.3 |
| | ResNet-50 | 50.9 | 77.7 |
| | ResNet-101 | 57.6 | 82.6 |
| | ResNet-152 | 58.3 | 82.9 |

**Table 8:** ImageNet-1k test *accuracy* (%) under $K$NN and linear classification on ResNet-50 encoder with *deeper*, MoCo-V2/SWAV pre-trained teacher architectures. ✗ denotes MoCo-V2 self-supervised learning baselines before distillation. * indicates using a stronger teacher encoder pre-trained by SWAV with additional small-patches during distillation.

| Teac.  Stud. | Epoch | $K$NN | ResNet-50 Top-1 | Top-5 |
|---|---|---|---|---|
| ✗ | 200 | 46.1 | 67.4 | 87.8 |
| ResNet-50 △ | 200 | 46.1 +0.0 | 67.5 +0.1 | 87.8 +0.0 |
| ResNet-101 △ | 200 | 52.3 +6.2 | 69.1 +1.7 | 88.7 +0.9 |
| ResNet-152 △ | 200 | 53.2 +7.1 | 70.4 +3.0 | 90.5 +2.7 |
| ResNet-50$_{\times 2}$* △ | 800 | 59.0 +12.9 | 74.3 +6.9 | 92.2 +4.4 |

of labeled ImageNet-1k dataset (1% and 10%), and evaluate it on the whole test split. The fraction of labeled dataset is constructed in a class-balanced way, with roughly 12 and 128 images per class*. We use SGD optimizer and set initial learning rate to be 30 with a multiplier = BatchSize/256 without weight decaying for 100 epochs. We use the step-wise scheduler for the learning rate updating with 5 warm-up epochs, and the learning rate is reduced by 10 at 60 and 80 epochs. On smaller architectures like EfficientNet and MobileNet, we reduce the initial learning rate to 3. During training, the image is center-cropped to the size of 224×224 with just Random Horizontal Flip as the data augmentation. For testing, we first resize the image to 256×256 and use the center cropped 224×224 for pre-processing. In Table 8, we show the distillation results on a larger encoder (ResNet-550) when using different teacher networks.

**Transfer Learning.** We test the transferability of the representations learned from self-supervised distillation by conducting the linear evaluations using offline features on several other datasets. Specifically, a single layer logistic classifier is trained following (Chen et al., 2020a; Grill et al., 2020) using SGD optimizer without weight decay and momentum parameter at 0.9. We use CIFAR-10, CIFAR-100 (Krizhevsky et al., 2009) and SUN-397 (Xiao et al., 2010) as our testing beds.

CIFAR: As the size for CIFAR dataset is 32×32, we resize all images to 224×224 pixels along the shorter side using bicubic resampling method, followed by a center crop operation. We set the learning rate at 1e-3 constantly and train it for 120 epochs. The hyper-parameters are searched using 10 fold cross-validation on the train split and report its final top-1 accuracy on the test split.

---

*The full image ids for semi-supervised evaluation on ImageNet-1k can be found at https://github.com/google-research/simclr/tree/master/imagenet_subsets.

**Table 9:** Object detection and instance segmentation fine-tuned on VOC07: bounding-box AP ($AP^{bb}$) and mask AP ($AP^{mk}$) evaluated on VOC07-val. The first row shows the baseline from MoCo-v2 backbones without distillation.

| Student | Teacher | VOC Object Detection | | |
| --- | --- | --- | --- | --- |
| | | $AP^{bb}$ | $AP^{bb}_{50}$ | $AP^{bb}_{75}$ |
| ResNet-34 | ✗ | 53.6 | 79.1 | 58.7 |
| | ResNet-50 | 53.7 (+0.1) | 79.4 (+0.3) | 59.2 (+0.5) |
| | ResNet-101 | 54.1 (+0.5) | 79.8 (+0.7) | 59.1 (+0.4) |
| | ResNet-152 | 54.4 (+0.8) | 80.1 (+1.0) | 59.9 (+1.2) |
| ResNet-50 | ✗ | 57.0 | 82.4 | 63.6 |
| | ResNet-50 | 57.0 (+0.0) | 82.4 (+0.0) | 63.6 (+0.0) |
| | ResNet-101 | 57.1 (+0.1) | 82.8 (+0.4) | 63.8 (+0.2) |
| | ResNet-152 | 57.3 (+0.3) | 82.8 (+0.4) | 63.9 (+0.3) |

**Table 10:** Object detection and instance segmentation fine-tuned on COCO: bounding-box AP ($AP^{bb}$) and mask AP ($AP^{mk}$) evaluated on COCO-val2017. The first several rows show the baselines from unsupervised backbones without distillation.

| Student | Teacher | Object Detection | | | Instance Segmentation | | |
| --- | --- | --- | --- | --- | --- | --- | --- |
| | | $AP^{bb}$ | $AP^{bb}_{50}$ | $AP^{bb}_{75}$ | $AP^{mk}$ | $AP^{mk}_{50}$ | $AP^{mk}_{75}$ |
| ResNet34 | ✗ | 38.1 | 56.8 | 40.7 | 33.0 | 53.2 | 35.3 |
| | ResNet50 | 38.4 (+0.3) | 57.0 (+0.2) | 41.0 (+0.3) | 33.3 (+0.3) | 53.6 (+0.4) | 35.4 (+0.1) |
| | ResNet101 | 38.5 (+0.4) | 57.3 (+0.5) | 41.4 (+0.7) | 33.6 (+0.6) | 54.1 (+0.9) | 35.6 (+0.3) |
| | ResNet152 | 38.4 (+0.3) | 57.0 (+0.2) | 41.0 (+0.3) | 33.3 (+0.3) | 53.7 (+0.5) | 35.3 (+0.0) |

SUN-397: We further extend our transferring evaluation to the scene dataset SUN-397 for a more diverse testing. The official dataset specifies 10 different train/test splits, with each contains 50 images per category covering 397 different scenes. We follow (Chen et al., 2020a; Grill et al., 2020) and use the first train/test split. For the validation set, we randomly pick 10 images (yielding 20% of the dataset), with identical optimizer parameters as CIFAR.

**Object Detection and Instance Segmentation.** As indicated by (He et al., 2020), features produced by self-supervised pre-training have divergent distributions in downstream tasks, thus resulting the supervised pre-training picked hyper-parameters not applicable. To relieve this, He et al. (2020) uses feature normalization during the fine-tuning phase and train the BN layers. Different from previous transferring and linear evaluations where we exploit only offline features, model for detection and segmentation is trained with all parameters tuned. For this reason, annotations on COCO for segmentation gives much higher influence for the backbone model than the VOC dataset (see Table 9), and gives an offset to the pre-training difference (see Table 10). Thus, this makes the performance boosting by pre-training less obvious, and leads to trivial AP differences before and after distillation.

Object Detection on PASCAL VOC-07: We train a C4 (He et al., 2017) based Faster R-CNN (Ren et al., 2015) as the detector with different ResNet architectures (ResNet-18, ResNet-34 and ResNet-50) for evaluating the transferability of features for object detection tasks. We use Detectron2 (Wu et al., 2019) for the implementations. We train our detector for 48k iterations with a batch size of 32 (8 images per GPU). The base learning rate is set to 0.01 with 200 warm-up iterations. We set the scale of images for training as [400, 800] and 800 at inference.

Object Detection and Segmentation on COCO: We use Mask R-CNN (He et al., 2017) with the C4 backbone for the object detection and instance segmentation task on COCO dataset, with 2× schedule. Similar to the VOC detection, we tune the BN layers and all parameters. The model is trained for 180k iterations with initial learning rate set to 0.02. We set the scale of images for training as [600, 800] and 800 at inference.

**Table 11:** Linear evaluations on ImageNet of ResNet-18 after distillation from the SWAV pre-trained ResNet-50 using either single view, cross-views, or small patch views.

| Method | Multi-View(s) | Top-1 Acc. | Top-5 Acc. |
|---|---|---|---|
| Identical-View | $1 \times 224$ | 61.7 | 84.2 |
| Cross-Views | $2 \times 224$ | 58.2 | 81.7 |
| Multi-Crops + Cross-Views | $1 \times 224 + 6 \times 96 \times 96$ | 61.9 | 84.4 |
| Multi-Crops + Identical-View | $1 \times 224 + 6 \times 96 \times 96$ | **62.6** | **84.8** |

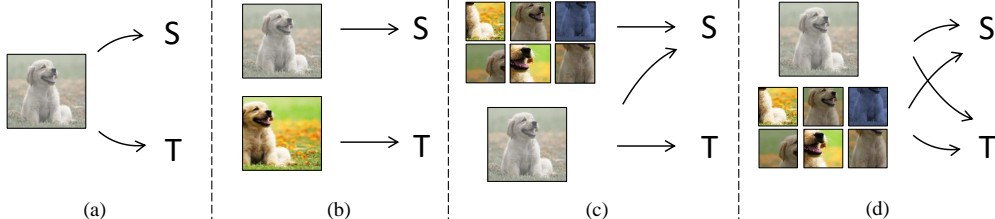

**Figure 6:** We experiment with different strategies of using views during distillation, which include: (a). Identical view for distillation. (b). Cross view distillation. (c). Large-small cross view distillation. (d). Large-small identical view distillation.

### A.3  SINGLE CROP V.S. MULTI-CROPS VIEW(S) FOR DISTILLATION

In contrary with most contrastive *SSL* methods where two different augmented views of an image are utilized as the positive samples (see Figure 6-a), SEED uses an identical view for each image (see Figure 6-b) during distillation and yields better performances, as is shown in Table. 11. In addition, we have also experimented with two strategies of using small patches. To be specific, we follow the set-up in SWAV (Caron et al., 2020), that 6 small patches of the size $96 \times 96$ are sampled at the scale of $(0.05, 0.14)$. Then, we apply the same augmentations as introduced previously as data pre-processing. Figure. 6-c shows the way that is similar in SWAV for small-patch learning, where both large and 6 small patches are fed into the student encoder, with the learning target ($\mathbf{z}^T$) to be the embedding of large view from the teacher encoder. Figure. 6-d is the strategy we use during distillation, that both views are fed into student and teacher to produce the embeddings for small-views ($\mathbf{z}_s^S$, $\mathbf{z}_s^T$) and large views ($\mathbf{z}_l^S$, $\mathbf{z}_l^T$). Based on that, the distillation is formulated separately on the small and large views. Notably, we maintain two independent queues for storing historical data samples for the large and small views.

### A.4  STRATEGIES FOR OTHER DISTILLATION METHODS

We compare the effect of distillation using different strategies with SEED.

*l2*-**Distance:** We train the student encoder by minimizing the squared *l2*-distance of representations from student ($\mathbf{z}_i^S$) and teacher ($\mathbf{z}_i^T$) for an identical view $\mathbf{x}_i$.

*K*-**Means:** We experiment with the *K*-Means clustering method to retrieve pseudo class labels for distillation. Specifically, we first extract offline image features using the *SSL* pre-trained Teacher network without any image augmentations. Based on this, we conduct our *K*-Means clustering with 4k and 16k unique centroids. Then the final centroids are used to produce pseudo labels for unlabelled instances. With that, we carry out the distillation by training the model on a classification task using the produced labels as the ground-truth. To avoid trivial solutions that the majority of images are assigned to a few clusters, we sample images based on a uniform distribution over pseudo-labels as clustering proceeds. We observe very close results when adjusting numbers of centroids.

**Online-Clustering:** With *K*-Means for pseudo-label generation training, it does not lead to satisfying results (51.0% on ResNet-18 with ResNet-50 as Teacher) as instances might have not been accurately categorized by limited frozen centroids. Similar to (Caron et al., 2018; Li et al., 2020), we resort to the "in-batch" and dynamical clustering to substitute the frozen *K*-Means method. We conduct

*K*-Means clustering within a batch and continuously update the centroid based on the teacher feature representation as distillation goes on. This alleviates the above problems and yields a substantial performance improvement on ResNet-18 to 56.4%.

**Binary Contrastive Loss:** We resort to CRD (Tian et al., 2019a) and adopt an *info-NCE* loss-alike training objective in unsupervised distillation tasks. Specifically, we treat representation features from Teacher and Student for instance $\mathbf{x}_i$ as positive pairs, and random instances from $\mathbf{D}$ as negative samples:

$$\hat{\theta}_S = \arg\min_{\theta_S} \sum_i^N \log h(\mathbf{z}_i^S, \mathbf{z}_i^T) + K \cdot [\log h(1 - h(\mathbf{z}_i^S, \mathbf{d}_j^T))], \tag{6}$$

where $\mathbf{d}_j^T \in \mathbf{D}$, $h(\cdot)$ is any family of functions that satisfy $h$: $\{\mathbf{z}, \mathbf{d}\} \to [0, 1]$, *e.g.*, cosine similarity.

## A.5 DISCUSSIONS ON SEED

Our proposed learning objective for SEED is composed of two goals, that is to align the encoding $\mathbf{z}^S$ by the student model with $\mathbf{z}^T$ produced by the teacher model; meanwhile, $\mathbf{z}^S$ also softly contrasts with random samples maintained in the $\mathbf{D}$. This can be formulated more directly as minimizing the *l2* distance of $\mathbf{z}^T$, $\mathbf{z}^S$, together with the cross-entropy computed using $\mathbf{D}$:

$$\mathcal{L} = \frac{1}{N} \sum_i^N \left\{ \lambda_a \cdot \left|\left| \mathbf{z}_i^T - \mathbf{z}_i^S \right|\right|_2 - \lambda_b \cdot \mathbf{p}^T(\mathbf{x}_i; \theta_T, \mathbf{D}) \cdot \log \mathbf{p}^S(\mathbf{x}_i; \theta_S, \mathbf{D}) \right\}$$
$$= \sum_i^N \left\{ -\lambda_a \cdot \mathbf{z}_i^T \cdot \mathbf{z}_i^S - \lambda_b \cdot \sum_j^K \frac{\exp(\mathbf{z}_i^T \cdot \mathbf{d}_j/\tau^T)}{\sum_{\mathbf{d} \sim \mathbf{D}} \exp(\mathbf{z}_i^T \cdot \mathbf{d}/\tau^T)} \cdot \log \frac{\exp(\mathbf{z}_i^S \cdot \mathbf{d}_j/\tau^S)}{\sum_{\mathbf{d} \sim \mathbf{D}} \exp(\mathbf{z}_i^S \cdot \mathbf{d}/\tau^S)} \right\}, \tag{7}$$

Directly optimizing *Eq. 7* can lead to apparent difficulty in searching optimal hyper-parameters $(\lambda_a, \lambda_b, \tau^T$ and $\tau^S)$. Our proposed objective on $\mathbf{D}^+$ indeed is an approximated upper-bound of the above objectiveness however much simplified:

$$\mathcal{L}_{\text{SEED}} = \frac{1}{N} \sum_i^N -\mathbf{p}^T(\mathbf{x}_i; \theta_T, \mathbf{D}^+) \cdot \log \mathbf{p}^S(\mathbf{x}_i; \theta_S, \mathbf{D}^+)$$
$$= \sum_i^N \sum_j^{K+1} - \underbrace{\frac{\exp(\mathbf{z}_i^T \cdot \mathbf{d}_j/\tau^T)}{\sum_{\mathbf{d} \sim \mathbf{D}^+} \exp(\mathbf{z}_i^T \cdot \mathbf{d}/\tau^T)}}_{\mathbf{w}_j^i} \cdot \log \frac{\exp(\mathbf{z}_i^S \cdot \mathbf{d}_j/\tau^S)}{\sum_{\mathbf{d} \sim \mathbf{D}^+} \exp(\mathbf{z}_i^S \cdot \mathbf{d}/\tau^S)}, \tag{8}$$

where we let $\mathbf{w}_j^i$ denote the weighting term regulated under $\tau^T$. Since the $(K+1)$th element in $\mathbf{D}^+$ is our supplemented vector $\mathbf{z}_i^T$, the above objective can be expanded into:

$$\mathcal{L}_{\text{SEED}} = \frac{1}{N} \sum_i^N \left\{ \mathbf{w}_{K+1}^i \cdot \left( -\mathbf{z}_i^S \cdot \mathbf{z}_i^T/\tau^S + \log \sum_{\mathbf{d} \sim \mathbf{D}^+} \exp(\mathbf{z}_i^S \cdot \mathbf{d}/\tau^S) \right) \right.$$
$$\left. + \sum_{j=1}^K \mathbf{w}_j^i \cdot \left( -\mathbf{z}_i^S \cdot \mathbf{d}_j/\tau^S + \log \sum_{\mathbf{d} \sim \mathbf{D}^+} \exp(\mathbf{z}_i^S \cdot \mathbf{d}/\tau^S) \right) \right\} \tag{9}$$

Note that the LSE term in the first line is strictly non-negative as the range of inner product for $\mathbf{z}^S$ and $\mathbf{d}$ lies between $\left[-1, +1\right]$:

$$\text{LSE}(\mathbf{D}^+, \mathbf{z}_i^S) \geq \log\left(M \cdot \exp(-1/\tau^S)\right) = \log\left(M \cdot \exp(-5)\right) > 0, \tag{10}$$

where $M$ denotes the cardinality of the maintained queue $\mathbf{D}^+$ and is set to 65,536 in our experiment with $\tau^S = 0.2$ constantly. Meanwhile, the LSE term in the second line satisfies the following inequality:

$$\text{LSE}(\mathbf{D}^+, \mathbf{z}_i^S) \geq \text{LSE}(\mathbf{D}, \mathbf{z}_i^S). \tag{11}$$

Thus, this demonstrates that the objective for SEED as *Eq.* 8 is equivalent to minimizing a weakened upper-bound of *e.q.* 7:

$$
\begin{aligned}
\mathcal{L}_{\text{SEED}} &= \frac{1}{N} \sum_i^N -\mathbf{p}^T(\mathbf{x}_i; \theta_T, \mathbf{D}^+) \cdot \log \mathbf{p}^S(\mathbf{x}_i; \theta_S, \mathbf{D}^+) \\
&\geq \frac{1}{N} \sum_i^N \left\{ \mathbf{w}_{K+1}^i \cdot (-\mathbf{z}_i^S \cdot \mathbf{z}_i^T/\tau^S) + \sum_{j=1}^K \mathbf{w}_j^i \cdot \left( -\mathbf{z}_i^S \cdot \mathbf{d}_j/\tau^S + \log \sum_{\mathbf{d} \sim \mathbf{D}} \exp(\mathbf{z}_i^S \cdot \mathbf{d}/\tau^S)) \right) \right\} \\
&= \frac{1}{N} \sum_i^N \left\{ -\frac{\mathbf{w}_{K+1}^i}{\tau^S} \cdot \mathbf{z}_i^S \cdot \mathbf{z}_i^T - \mathbf{p}^T(\mathbf{x}_i; \theta_T, \mathbf{D}) \cdot \log \mathbf{p}^S(\mathbf{x}_i; \theta_S, \mathbf{D}) \right\}
\end{aligned}
\tag{12}
$$

This proves that our $\mathcal{L}_{\text{SEED}}$ directly relates to a more intuitive distillation formulation as *Eq.* 7 (*l2* + cross entropy loss), and it implicitly contains the objective of aligning and contrasting. However, our training objective is much simplified. During practice, we find by regulating $\tau^T$, both training losses produce equal results.

## A.6 DISCUSSION ON THE RELATIONSHIP OF SEED WITH INFO-NCE

The objective of distillation can be considered as a soft version of Info-NCE (Oord et al., 2018), with the only difference to be that SEED learns from the negative samples with probabilities instead of treating them all strictly as negative samples. To be more specific, following Info-NCE, the "hard" style contrastive distillation can be expressed as aligning with representations from the Teacher encoder and contrasting with all random instances:

$$
\hat{\theta}_S = \arg\min_{\theta_S} \mathcal{L}_{NCE} = \arg\min_{\theta_S} \sum_i^N -\log \frac{\exp(\mathbf{z}_i^T \cdot \mathbf{z}_i^S/\tau)}{\sum_{\mathbf{d} \sim \mathbf{D}} \exp(\mathbf{z}_i^S \cdot \mathbf{d}/\tau)}
\tag{13}
$$

which can be further deduced with two sub-terms consisting of positive sample alignment and contrasting with negative instances:

$$
\mathcal{L}_{NCE} = \sum_i^N \left\{ \underbrace{-\mathbf{z}_i^S \cdot \mathbf{z}_i^T/\tau}_{alignment} + \underbrace{\log \sum_{\mathbf{d} \sim \mathbf{D}} \exp(\mathbf{z}_i^S \cdot \mathbf{d}/\tau)}_{contrasting} \right\}.
\tag{14}
$$

Similarly, the objective of SEED can be dissembled into the weighted form of alignment and contrasting terms:

$$
\begin{aligned}
\mathcal{L}_{\text{SEED}} &= \frac{1}{N*M} \sum_i^N \sum_j^{K+1} -\frac{\exp(\mathbf{z}_i^T \cdot \mathbf{d}_j/\tau^T)}{\sum_{\mathbf{d} \sim \mathbf{D}^+} \exp(\mathbf{z}_i^T \cdot \mathbf{d}/\tau^T)} \cdot \log \frac{\exp(\mathbf{z}_i^S \cdot \mathbf{d}_j/\tau^S)}{\sum_{\mathbf{d} \sim \mathbf{D}^+} \exp(\mathbf{z}_i^S \cdot \mathbf{d})/\tau^S} \\
&= \frac{1}{N*M} \sum_i^N \sum_j^{K+1} \underbrace{\frac{\exp(\mathbf{z}_i^T \cdot \mathbf{d}_j/\tau^T)}{\sum_{\mathbf{d} \sim \mathbf{D}^+} \exp(\mathbf{z}_i^T \cdot \mathbf{d}/\tau^T)}}_{\mathbf{w}_j^i} \cdot (\underbrace{-\mathbf{z}_i^S \cdot \mathbf{z}_i^T/\tau^S}_{alignment} + \underbrace{\log \sum_{\mathbf{d} \sim \mathbf{D}} \exp(\mathbf{z}_i^S \cdot \mathbf{d}/\tau^S)}_{contrasting})),
\end{aligned}
\tag{15}
$$

where the normalization term can be considered as soft labels, $\mathbf{W}^i = \left[ \mathbf{w}_1^i \dots \mathbf{w}_{K+1}^i \right]$, which can weight the above loss as:

$$
\mathcal{L}_{\text{SEED}} = \frac{1}{N*M} \sum_i^N \sum_j^{K+1} \mathbf{w}_j^i \cdot \left\{ -\mathbf{z}_i^S \cdot \mathbf{z}_i^T/\tau^S + \log \sum_{\mathbf{d} \sim \mathbf{D}} \exp(\mathbf{z}_i^S \cdot \mathbf{d}/\tau^S)) \right\},
\tag{16}
$$

When tuning hyper-parameter $\tau^T$ towards 0, $\mathbf{W}^i$ can be altered into the format of one-hot vector with $\mathbf{w}_{K+1}^i = 1$, which is then degraded to the case of contrastive distillation as in equation 14. In practice, the choice of an optimal $\tau^T$ can be dataset-specific. We show that the higher $\tau^T$ (with labels be more 'soft') can actually yield better results on other datasets, *e.g.*, CIFAR-10 (Krizhevsky et al., 2009).

## A.7 COMPATIBILITY WITH SUPERVISED DISTILLATION

SEED conducts self-supervised distillation at the pre-training phase for the representation learning. However, we verify that SEED is compatible with traditional supervised distillation that happened during fine-tuning phrase at downstream, and can even produce better results. We begin with the *SSL* pre-training on a larger architecture (ResNet-152) using MoCo-V2 and train it for 200 epochs as the teacher network. As images in CIFAR-100 are in the size of $32 \times 32$, we modify the first *conv* layer in ResNet with kernel size = 3 and stride = 1.

We then compare the Top-1 accuracy of a smaller ResNet-18 on CIFAR-100 when using different distillation strategies when all parameters are trainable. First, we use SEED to pre-train ResNet-18 with Res-152 as the teacher model, and then evaluate in on the test split of CIFAR-100 using linear fine-tuning task. As we keep all parameters trainable during the fine-tuning phase, distillation on the pre-training only yields a trivial boost: 75.4% *v.s.* 75.2%. Then, we adopt the traditional distillation method, *e.g.*, (Hinton et al., 2015), to first fine-tune the ResNet-152 model, and then use its output class probability to facilitate the linear classification task on ResNet-18 in the fine-tuning phrase. This improves the linear classification accuracy on ResNet-18 to 76.0%. At the end, we initialize the ResNet-18 with our SEED pre-trained ResNet-18, and equip it with the supervised classification distillation during fine-tuning. With that, we find that the performance of ResNet-18 is further boosted to 78.1%. We can conclude that our SEED is compatible with traditional supervised distillation that mostly happened at downstream for specific tasks, *e.g.*, classification, object detection.

**Table 12:** CIFAR-100 Top-1 *Accuracy*(%) of ResNet-18 with (or without) distillation at different phase: self-supervised pre-training stage, and supervised classification fine-tuning. All backbone parameters of ResNet-18 are trainable in experiments.

| Pre-training Distill. | Fine-tuning Distill. | Top-1 Acc |
|:---:|:---:|:---:|
| ✗ | ✗ | 75.2 |
| ✓ | ✗ | 75.4 |
| ✗ | ✓ | 76.0 |
| ✓ | ✓ | 78.1 |

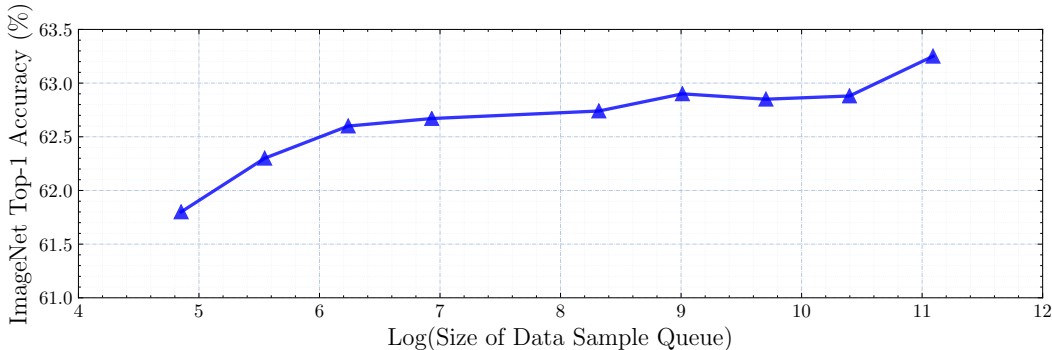

**Figure 7:** Linear evaluation *accuracy* (%) of distillation between ResNet-18 (as the Student) and ResNet-50 (as the Teacher) using different size of queue when LR=0.03 and weight decay=1e-6. Note the axis is the log(·) value of queue lengths.

## A.8 ADDITIONAL ABLATION STUDIES

We study effects of different hyper-parameters to distillation using a ResNet-18 (as Student) and a SWAV pre-trained ResNet-50 (as Teacher) with small patch views. In specific, we list the Top-1 Acc. on validation split of ImageNet-1k using different lengths of queue ($K$=128, 512, 1,024, 4,096, 8,192, 16,384, 65,536) in Figure. 7. With the increasing of random data samples, the distillation boosts the accuracy of learned representations, however within a limited range: +1.5 when the queue size is

**Table 13:** Linear evaluation *accuracy* (%) of distillation between ResNet-18 (as the Student) and ResNet-50 (as the Teacher) using different learning rates when the queue size is 65,536 and weight decay=1e-6.

| LR | Top-1 Acc. | Top-5 Acc. |
|------|------|------|
| 1 | 58.9 | 83.1 |
| 0.1 | 62.9 | 85.3 |
| 0.03 | **63.3** | **85.4** |
| 0.01 | 62.6 | 85.0 |

**Table 14:** Linear evaluation *accuracy* (%) of distillation between ResNet-18 (as the Student) and ResNet-50 (as the Teacher) using different weight decays when the queue size is 65,536 and LR=0.03.

| WD | Top-1 Acc. | Top-5 Acc. |
|------|------|------|
| 1e-2 | 11.8 | 27.7 |
| 1e-3 | 62.3 | 84.7 |
| 1e-4 | 61.9 | 84.4 |
| 1e-5 | 61.6 | 84.2 |
| 1e-6 | **63.3** | **85.4** |

65,536 compared with 256. Furthermore, Table. 13 and 14 summarize the linear evaluation accuracy under different learning rates and weight decays.

