# OpenReview forum: "SEED: Self-supervised Distillation For Visual Representation"
_ICLR.cc/2021/Conference — ICLR 2021 Poster_

### Official Review · AnonReviewer4 · 2020-10-22
**Interesting work and solid results with various network architecture**

**Rating:** 6
**Confidence:** 5

**Review:**

Summary:
The paper address the problem of knowledge distillation in self-supervised learning, where the representational knowledge from the larger model (i.e., teacher) is used to guide the learning of a smaller model (i.e, student). To achieve this, an instance queue is used to compute the similarity score between teacher model features and the feature of a given image, and the learning objective is to minimise the cross-entropy loss of the similarity between teacher and student models. The paper provides comprehensive empirical results to justify the efficacy of the proposed approach.

Justification of rating:
The paper provides a comprehensive evaluation of the proposed approach on various network architecture and downstream tasks. Overall, I feel this work does not have sufficient theoretical or algorithmic contributions. The key contribution is the idea of apply knowledge distillation for self-supervised learning.

Strengths:
+ This is the first work that addresses self-supervised learning (SSL) with knowledge distillation. It empirically shows SSL with a small model is challenging (consistent with finding from Chen et al., 2020a;b), and proposed a technique (SEED) to transfer knowledge.
+ This paper provides comprehensive experiment on image classification task (self-supervised, semi-supervised and supervised), object detection/segmentation, domain transfer, as well as provide ablation studies on various model architecture and parameters. The results show knowledge distillation is effective for self-supervised learning.

Weakness:
- The core novelty of this work is the idea of conduct knowledge distillation in self-supervised learning. The key weakness is that the knowledge distillation approach and the instance queue approach are previously proposed and known to the research community. This work empirically shows how it can be combined for the task on hand.

Minor comments:
- In section 3.2, before eqn (2), it is best to change “the similarity score between $x_i$ and $\vec{d}_j$’s” to "the similarity between the extracted (teacher/student) feature $z_i$ and $\vec{d}_j$’s”. This is to avoid confusion as one might wonder how can the similarity between the input image and the $\vec{d}_j$ be computed.
- Please move the table’s caption to the top of the table.

---

> ### Author Response · Authors · 2020-11-18
> **We appreciate the reviewer for his/her comments.**
>
> We appreciate the reviewer for his/her comments.
>
> **Q13**: a notation change in equation (2).
>
> **A13**: We thank the reviewer for his/her careful reading, and we will address the confusion in our updated version accordingly.
>
> **Q14**: Please move the table’s caption to the top of the table.
>
> **A14**: We will address the caption issue properly in our updated version, and we thank the reviewer for pointing this out.

---

### Official Review · AnonReviewer2 · 2020-10-24
**It is interesting to see distillation applied to self-supervised learning. Experimental results are comprehensive.**

**Rating:** 7
**Confidence:** 5

**Review:**

The paper proposes to distill knowledge from large teacher networks to small student networks in self-supervised learning. Experimental results show significant improvements on small networks.

Concerns:

- During the self-supervised distillation phase, it is not clear to me only distillation loss is applied or self-supervised learning loss is combined with distillation loss for learning student network. If only distillation loss is applied, does it make sense to train a student network using both self-supervised learning loss and distillation loss, such as MoCo-v2 used in most experiments?
- The caption of Figure 3 is confusing, it would be good to explain it more clearly.
- It seems that improvements on object detection and instance segmentation (Table 2) are relatively small compared to other experiments, are there any explanations? Could it be possible to use smaller student networks in this experiment as well?
- In the experiment of different sizes of sample queue, does it mean that the larger the better? What is the intuition behind it?
- Strong data augmentation is needed for most of self-supervised methods, but normally for distillation, it is not common to use very strong data augmentation, why does the paper decide to use the same data augmentation for both self-supervised learning and distillation learning?
- It is quite similar to a very recent paper as shown in the following, it would be good to discuss the differences in the paper.

@inproceedings{koohpayegani2020compress,
  title={CompRess: Self-Supervised Learning by Compressing Representations},
  author={Koohpayegani, Soroush Abbasi and Tejankar, Ajinkya and Pirsiavash, Hamed},
  booktitle={NeurIPS},
  year={2020}
}

---

> ### Author Response · Authors · 2020-11-18
> **We thank the reviewer for his/her constructive comments.**
>
> We thank the reviewer for his/her constructive comments.
>
> **Q6**: It is not clear to me only distillation loss is applied or self-supervised learning loss is combined with distillation loss for learning student network.
>
> **A6**: We conduct experiments studying the effect of the original SSL (MoCo-V2) loss and find it does not bring additional benefits on top of self-supervised distillation loss.
> Specifically, we set-up our experiment using a ResNet-18 distill from a MoCo-V2 pre-trained ResNet-50 for 200 epochs. We maintain two individual instance queues for SEED distillation and MoCo-V2 SSL on Student respectively. After evaluation, we find close results from these two strategies, SEED achieves 57.9%  (Top-1 linear Acc.), while SEED + MoCo-V2 achieves 57.6%.
> This implies that the loss of SEED can to a large extent cover the original SSL loss, and it is not necessary to add SSL loss. Compared to SEED, SSL loss does not leverage any extra information as the optimization target. As representations from the Teacher have been well pre-trained by the SSL loss, representations from it already fit the principle of instance discrimination. SEED proposes to mimic the Teacher representations, which has the same effect as optimizing SSL objectives.
>
> **Q7**: The caption of Figure 3 is confusing, it would be good to explain it more clearly.
>
> **A7**: Figure 3 shows the semi-supervised linear evaluations on ImageNet, using 1% (red line), and 10% (blue line) of the labeled images, and compares with the full supervision linear evaluations (green line). This experiment shows that the representations are robust when only partial labels are available for fine-tuning. We will make the explanations more clear in our updated version.
>
> **Q8**: Improvements in object detection are small compared to other experiments. And is it possible to use smaller student networks here?
>
> **A8**: Several reasons explain the limited improvement in detection/segmentation tasks:
> - Experimental settings. We follow MoCo and make parameters on these two tasks tuneable during fine-tuning with fairly long training epochs (schedule X 2). This makes the model initialization less important when the fine-tuning schedule is long enough or with a large number of annotations. See more discussion in A1.
> - The domain gap between pre-training and fine-tuning can yield a negative impact resulting in a diminishing gain.
> Table 2 in the main paper actually shows the results of a relatively small detector (using ResNet18 as the backbone). We additionally show the results of larger architectures (ResNet34 and ResNet50) in the Appendix. We will try even smaller networks in our future version.
>
> **Q9**: In the experiment of different sizes of sample queue, does it mean that the larger the better? What is the intuition behind it?
>
> **A9**: We studied the effect of larger instance queues in Sec. A.1.8, Appendix, where we compare the different sizes of the queue from 128 to 65,536. Empirically, we find that the use of a larger queue can bring only marginal performance improvement: 63.3% for 65,536 and 61.7% for 128. Further enlarging the instance queue does not obviously contribute to the results.
> Intuitively, more samples in a queue mean more contrastiveness to help the representation to be discriminative to more negative samples while maintaining similarity with the positive sample. However, extra negative samples could also reduce the importance of being similar to the positive sample to some extent. Thus, an even larger queue does bring improvement, however is limited.
>
>
> **Q10**: Why does the paper decide to use the same data augmentation for SSL/SEED?
>
> **A10**: Intuitively, stronger augmentation provides the training with more diverse images, yielding representations with better generalization. From another aspect, SSL and SEED have much longer training epochs than the supervised training. In such a case, weaker augmentation can make the network overfit to the less-augmented image.
> We find that it is important to maintain a consistent data distribution between the SSL pre-training and distillation as SEED has a similar optimization objective to SSL.
> For these reasons, we choose to use identical augmentations as SSL pre-training, which is different from traditional distillation settings. We will attach additional ablations to study the effects of various augmentations in the future.
>
> **Q11**: Add related work of CompRess.
>
> **A11**: We thank the reviewer for bringing up CompRess as a concurrent work for us. We will add this as related work in our updated version.

---

### Official Review · AnonReviewer1 · 2020-10-29
**Simple approach for SSL knowledge distillation with good results**

**Rating:** 7
**Confidence:** 4

**Review:**

Summary: This paper proposes a  knowledge distillation (KD) approach for self-supervised learning (SSL) with small neural network models. The authors first observe that the state-of-the-art contrastive learning-based SSL does not obtain good performance on small models, due  to the larger  model capacity required for instance discrimination. To tackle this problem, they propose a SEED, a  KD method where the smaller student model learns to mimic its larger teacher model’s similarity distribution between an instance and its augmented views, using a cross-entropy based objective. The authors perform various experiments to show that -- 1)  SEED obtains substantial improvement in SSL-based imagenet classification performance for small models as compared to SSL training without SEED, 2) the performance gains are also substantial for transfer learning on other classification tasks, 3) the performance gains are smaller for downstream tasks of object detection and instance segmentation, with performance gains reducing for the larger COCO dataset, as compared to VOC, 4) SEED is robust to choice of SSL method, and performs better than other KD approaches.

Strengths: The paper is clearly written and well-organized. The SEED approach  is well-motivated and sensible. Experimental validation and the ablation studies are quite thorough. Performance gains on classification tasks are substantial. The method is simple to implement.

Weaknesses: 1. Performance gains on downstream tasks of detection and instance segmentation are much lower -- how would the authors propose to improve these? 2. If the primary goal is to improve SSL performance on small models, I would have liked to see more analysis on how different design choices of setting up contrastive learning affect model performance and if these could aid performance improvement, in addition to knowledge distillation.

Questions and suggestions: 1. Adding fully-supervised baselines for small models in table 1 will be useful in understanding the gap between full supervision and SSL for these models. 2. In figure 3, does 100% (green line) represent the student network trained with 100% of labeled imagenet supervised data? It is hard to interpret what these numbers represent. 3. Minor point: Some citations, which should not be in parentheses, are in parentheses (e.g., Romero et al. page 8). Please fix this in the revision.

---

> ### Author Response · Authors · 2020-11-18
> **We thank the reviewer for his/her comments and suggestions.**
>
> We thank the reviewer for his/her comments and suggestions.
>
> **Q1**: Performance gains on downstream tasks of detection and instance segmentation are much lower. How would the authors propose to improve these?
>
> **A1**: The detection/segmentation tasks are challenging for several reasons:
> * Following MoCo, we make the parameters of the backbone from self-supervised pre-training/distillation tunable for detection/segmentation fine-tuning. Accordingly, this makes the model initialization less important when the fine-tuning schedule is long enough or with a large number of annotations: e.g., performance gain on COCO is much less obvious than the gain on the VOC dataset. This trend also aligns with the observation in MoCo where limited improvement is obtained when trained under longer (schedule x 2) epochs on COCO: MoCo gains only +0.7 on AP_mk/bb, COCO dataset.
> * Domain gap between ImageNet and COCO. As the distillation is carried out on ImageNet-1k Dataset, we conjecture that the domain gap between pre-training and fine-tuning can yield a negative impact resulting in the diminishing gain.
>
> We think this issue is indeed an interesting open challenge for future work: how can the generic SSL learned representations consistently perform well when transferred to other tasks/domains. We list several aspects that are potentially useful for addressing this:
> - The inconsistent optimization objectives for pre-training and fine-tuning. Current SSL supervision is solely about image-level discrimination without location information, i.e., image-level classification during SSL, versus object-level localization in fine-tuning. Thus, to design a task-specific objective that incorporates more location information for pre-training/distillation is necessary.
> - Conduct the self-supervised distillation at downstream. This might be helpful to mitigate the domain gaps.
>
>
> **Q2**: Reviewer suggests to show more analysis on how different design choices of contrastive learning and settings affect performances of small models, and if these could aid performance improvement in addition to knowledge distillation.
>
> **A2**: We study how several different settings in contrastive learning affect SSL performances on small models. We find that expanding deeper MLP projection heads in small models improves the SSL performances (see Sec. A.1.2 of Supplementary materials). Specifically, SSL using MoCo-v2 on EfficientNet-b0 only yields a 39.1% linear evaluation Top-1 Acc. on ImageNet, which is increased to 42.2% after using a deeper MLP head. Compared to the improvement brought by SEED (over 25% Top-1 Acc. gain on EfficientNet), additional variations with other factors like larger memory queue, stronger augmentation do not bring obvious benefits.
> Recent concurrent work ‘CompRess’ as suggested by the reviewer also finds similar results, that small model suffers from self-supervised training. Notably, by using another state-of-the-art SSL method (i.e., SimCLR), small models still perform rather worse: AlexNet achieves only 46% Top-1 Acc., ResNet18 only achieves 51% Top-1 Acc. after SSL pre-training, which aligns with our observation using MoCo-v2: 52.5% Top-1 Acc using ResNet18.
>
>
>
> **Q3**: Adding fully-supervised baselines for small models in table 1 will be useful in understanding the gap between full supervision and SSL for these models.
>
> **A3**: We thank the reviewer for pointing out this, and we now have added the fully-supervised baselines for smaller models in Table 1.
>
> **Q4**: In figure 3, does 100% (green line) represent the student network trained with 100% of labeled imagenet supervised data? It is hard to interpret what these numbers represent.
>
> **A4**: Yes, the green line represents classification results trained with 100% labeled on ImageNet. We have added additional explanations in the caption to make it more clear.
>
>
> **Q5**: Minor point: Some citations, which should not be in parentheses, are in parentheses (e.g., Romero et al. page 8). Please fix this in the revision.
>
> **A5**: We have addressed this accordingly in the updated version.
>
> [1] Kaiming He, Haoqi Fan, Yuxin Wu, Saining Xie, and Ross Girshick. Momentum contrast for unsupervised visual representation learning. In Proceedings of the IEEE/CVF Conference on Computer Vision and Pattern Recognition, pp. 9729–9738, 2020.2
> [2] CompRess: Self-Supervised Learning by Compressing Representations, Koohpayegani Soroush Abbasi, Tejankar Ajinkya and Pirsiavash Hamed.

---

### Decision · Program_Chairs · 2021-01-07
**Final Decision**

**Decision:**

Accept (Poster)

**Comment:**

There is definite consensus on this paper, with all reviewers expressing very favorable opinions. The author responses are very well articulated and address the main concerns expressed by the reviewers. The paper is very well-written and the ablation study well-executed. Some recent related work was missed in the original submission, but this was adequately addressed in rebuttal. The proposed approach is novel technique for feature representation learning. The clarifications to the manuscript and the new analyses are especially appreciated.